# Pfizer-BioNTech COVID-19 Vaccine in Gynecologic Oncology Patients: A Prospective Cohort Study

**DOI:** 10.3390/vaccines10010012

**Published:** 2021-12-23

**Authors:** Innocenza Palaia, Giuseppe Caruso, Violante Di Donato, Annarita Vestri, Anna Napoli, Giorgia Perniola, Matteo Casinelli, Danilo Alunni Fegatelli, Roberta Campagna, Federica Tomao, Debora D’Aniello, Guido Antonelli, Ludovico Muzii

**Affiliations:** 1Department of Maternal and Child Health and Urological Sciences, Policlinico Umberto I, Sapienza University, 00161 Rome, Italy; innocenza.palaia@uniroma1.it (I.P.); violante.didonato@uniroma1.it (V.D.D.); giorgia.perniola@uniroma1.it (G.P.); federica.tomao@uniroma1.it (F.T.); debora.daniello@uniroma1.it (D.D.); ludovico.muzii@uniroma1.it (L.M.); 2Department of Public Health and Infectious Diseases, Policlinico Umberto I, Sapienza University, 00161 Rome, Italy; annarita.vestri@uniroma1.it (A.V.); danilo.alunnifegatelli@uniroma1.it (D.A.F.); 3Department of Molecular Medicine, Policlinico Umberto I, Sapienza University, 00161 Rome, Italy; anna.napoli@uniroma1.it (A.N.); matteo.casinelli@uniroma1.it (M.C.); roberta.campagna@uniroma1.it (R.C.); guido.antonelli@uniroma1.it (G.A.)

**Keywords:** COVID-19, SARS-CoV-2, cancer, gynecologic oncology, vaccine, chemotherapy

## Abstract

Objective: To evaluate the safety and immunogenicity of the Pfizer-BioNTech COVID-19 vaccine in gynecologic oncology patients under chemotherapy. Methods: A prospective cohort study including gynecologic oncology women who were under chemotherapy or had completed it within 6 months at the time of the study. All patients received a two-dose schedule of the Pfizer-BioNTech COVID-19 vaccine. Results were compared with a control group of healthy women vaccinated in the same period. Results: Overall, 44 oncologic patients with a mean age of 61.3 ± 10.7 years were enrolled: 28 (63.6%) had ovarian cancer, 9 (20.4%) endometrial, and 7 (16%) cervical. The IgG antibody titer after 1 month from vaccination was low in 9 (20.5%) patients, moderate in 21 (47.7%), and high in 14 (31.8%). The 3-month titer was null in 2 (4.5%) patients, low in 26 (59.1%), moderate in 13 (29.5%), and high in 3 (6.8%). Patients ≥ 50 years reported lower 1-month (*p* = 0.018) and 3-month (*p* = 0.004) titers compared with <50 years. Patients with BMI < 30 kg/m^2^ had a higher 1-month titer compared with BMI ≥ 30 kg/m^2^ (*p* = 0.016). Compared with healthy women (*n* = 44), oncologic patients showed a lower 3-month titer (*p* < 0.001). None of the patients experienced serious adverse effects. Conclusions: The COVID-19 vaccine was safe and immunogenic in gynecologic oncology patients under chemotherapy. Serological monitoring and further vaccine shots should be considered to boost protection.

## 1. Introduction

The global cancer burden using the updated GLOBOCAN database estimated 19.3 million new cancer cases and nearly 10.0 million cancer deaths in 2020 worldwide [1]. Managing this elevated number of cancer patients during the COVID-19 (SARS-CoV-2) pandemic represents a real clinical challenge [2,3]. Since the beginning of the global COVID-19 spread in early 2020, cancer patients have been considered as a particularly vulnerable subset of the population, being at increased risk of severe COVID-19 infection, complications, and death [4,5,6,7].

Practicing preventative measures and getting vaccinated are the only valid strategies during this COVID-19 pandemic [8,9,10]. International guidelines now recommend that most people with cancer or a history of cancer should undergo COVID-19 vaccination and be positioned at high prioritization, based on the assumption that the benefits outweigh the risks [6,8,9,10,11,12,13,14,15,16]. However, till today there have not been enough data to prove COVID-19 vaccine safety and immunogenicity specifically in cancer patients [17,18]. The degree and durability of immunogenicity in patients with cancer are still unknown and unexplored and may be modulated by several factors including age, type and extent of disease, cancer treatment, and comorbidities [11,13,16]. The main concern about getting the vaccine is not so much related to its safety but rather to its immunogenicity, considering that either the disease itself (e.g., leukemias, lymphomas) or the treatment (e.g., chemotherapy, radiotherapy, immunotherapy, surgery) might weaken the immune system, thus decreasing seroconversion rates [10,19]. Moreover, there are no data on whether the immune response to the SARS-CoV-2 is influenced by active cancer and ongoing or recent cancer treatments.

The present study aimed to evaluate the safety and immunogenicity of the Pfizer-BioNTech COVID-19 vaccine in gynecologic oncology patients under chemotherapy treatment and to investigate the range of potential risk factors modulating the immunogenicity.

## 2. Materials and Methods

This was a single-center, observational prospective cohort study including gynecologic oncology women who received a two-dose schedule (21 days apart) of the Pfizer-BioNTech COVID-19 vaccine [20] while undergoing chemotherapy at the Department of Gynecologic Oncology of Umberto I Hospital in Rome. The first dose was administered on 19 March 2021 and the second one on 9 April 2021. Both doses were administered at least 7–14 days apart from chemotherapy infusions. An antibody titer blood test was performed one and three months after the second dose to determine the presence and measure the levels of anti-SARS-CoV-2 IgG antibodies. All results were compared with a control group of healthy female healthcare workers from our institution vaccinated in the same period. Local and systemic events both after the first and second dose of vaccination in both groups (cancer patients and healthy controls) were compared over one week. Patients provided written informed consent and the procedures were in accordance with the Helsinki declaration.

Key inclusion criteria included: (a) women aged 18 years and older; (b) histologically confirmed diagnosis of primary ovarian, endometrial, or cervical cancer; (c) patients under chemotherapy treatment or who had completed chemotherapy within the prior 6 months; (d) no clinical contraindications to COVID-19 vaccination (e.g., severe allergic reaction to a previous dose or a known allergy to a component of the vaccine); (e) negative baseline SARS-CoV-2 IgG antibodies before vaccination; (f) written informed consent. All patients were submitted to a nasopharyngeal molecular (RT-PCR) swab testing, with negative results, before receiving the COVID-19 vaccine.

The LIAISON^®^ SARS-CoV-2 TrimericS IgG assay was the method used to detect anti-trimeric S spike protein IgG antibodies [21,22]. LIAISON^®^ SARS-CoV-2 TrimericS IgG assay is a second-generation chemiluminescence immunoassay (CLIA) for the quantitative detection of anti-trimeric spike protein-specific IgG antibodies to SARS-CoV-2 in human serum or plasma samples. The clinical laboratory IgG titers were expressed in AU/mL; values inferior to 300 were categorized as low, from 300 to 800 as moderate, and >800 as high titer.

Progressive disease at the time of vaccination was defined according to the RECIST 1.1 criteria [23] (at least a 20% increase in the sum of the largest diameter of target lesions, taking as reference the smallest sum recorded since the treatment started, or the appearance of one or more new lesions) or predicted by an increase in CA125 levels.

Demographic and clinical data were collected from electronic medical records and entered into an Excel database. Quantitative variables are expressed as mean (standard deviation, SD) and median (interquartile range, IQR). Categorical variables are expressed as counts and percentages. Comparisons between groups were made using Kruskal–Wallis test or chi-square test. A statistical level of 0.05 was used for all the analyses. Statistical software R version 4.0.4 was used for all the analyses.

## 3. Results

Overall, 44 patients were enrolled in the study. Demographic, clinical, and oncological data are detailed in Table 1.

Briefly, the mean age was 61.3 ± 10.7 years, and the mean BMI was 25.5 ± 5.0 kg/m^2^. Comorbidities were reported in 33 patients (75%), including thyroid disorders (36.4%), cardiovascular disease (22.7%), and respiratory disease (6.8%). The primary cancer was ovarian in 28 patients (63.6%), endometrial in 9 (20.4%), and cervical in 7 patients (16%). The FIGO (International Federation of Gynecology and Obstetrics) stage was I in 12 patients (27.3%), II in 8 (18.2%), III in 20 (45.5%), and IV in 4 patients (9.1%). Chemotherapy was in progress at the time of vaccination in 30 (68.2%) patients, while 14 (31.8%) had completed therapy within the preceding 6 months. The disease was progressing at the time of vaccination in 23 (52.3%) patients, while it was stable or in (partial or complete) remission in the remaining patients. There were no statistically significant differences in terms of demographic data between cancer patients and controls.

Table 2 summarizes the side effects after the second dose of the Pfizer-BioNTech COVID-19 vaccine.

None of the patients experienced serious adverse effects after vaccination. Pain at the injection site was the most commonly reported side effect both after the first and the second dose. No statistically significant differences in terms of post-vaccine adverse events were reported between cancer patients and healthy controls.

The IgG antibody titer after 1 month from vaccination was low in 9 (20.5%) patients, moderate in 21 (47.7%), and high in 14 (31.8%). The IgG antibody titer after 3 months from vaccination was null in 2 (4.5%) patients, low in 26 (59.1%), moderate in 13 (29.5%), and high in 3 (6.8%). Table 3 describes the relationship between the antibody titers and several variables, including age, weight, comorbidities, progressive disease, FIGO stage, and chemotherapy in progress during vaccination.

Table 4 shows that patients aged ≥50 years reported a lower antibody titer both after 1 month (*p* = 0.018) and 3 months (*p* = 0.004) from vaccination compared with those with less than 50 years of age.

Moreover, it shows that patients with BMI <30 kg/m^2^ had a higher 1-month antibody titer compared with those with BMI ≥30 kg/m^2^ (*p* = 0.016). No statistically significant differences were reported between the antibody titer and the other analyzed variables, i.e., progressive disease, comorbidities, FIGO stage, and chemotherapy in progress.

Finally, Table 5 compares the 1-month and 3-month antibody titers between oncologic patients and healthy vaccinated women.

Although no significant difference in terms of antibody titer was reported after 1 month from vaccination, the titer after 3 months was lower in oncologic patients compared with healthy women (*p* < 0.001).

## 4. Discussion

In the context of the ongoing global COVID-19 pandemic, caused by severe acute respiratory syndrome coronavirus 2 (SARS-CoV-2), the preventive role of vaccines becomes even more important for immunocompromised individuals, such as cancer patients. However, there is only scant evidence specifically addressing the immunogenicity and safety of available vaccines in cancer patients since immunocompromised individuals were excluded from initial registration trials. Moreover, while there are sufficient data to consider the vaccine safe even in the cancer subpopulation, less is known about the degree and durability of immunogenicity, either in the general population or specifically in cancer patients [24]. The serological trending over time has not been explored, and there is no clear antibody cutoff that has been demonstrated to guarantee protection against SARS-CoV-2 infection. In addition, the term cancer does not refer to a single disease but instead gathers several and different types of histological subtypes of diseases, each presenting specific characteristics and molecular profiles.

To our knowledge, this is the first study specifically addressing the safety and immunogenicity of the Pfizer-BioNTech COVID-19 vaccine in gynecologic oncology women under chemotherapy treatment compared with a control group. Recently, Forster et al. published a German study reporting that COVID-19 vaccination was well-tolerated by patients with breast and gynecological cancer undergoing systemic cancer therapy [18]. Our data show that the vaccine was well tolerated by cancer patients as well as by control healthy women; indeed, only mild adverse effects were reported. The most common adverse effects were pain at the injection site, asthenia, headaches, and diffuse myalgias, which were easily managed. This is in line with the encouraging published data suggesting that the vaccine is safe even in the fragile cancer subpopulation [13,14,15,16]. All our patients experienced an adequate seroconversion, and none of them had COVID-19 infection during the study.

Few studies compared the antibody titer between cancer patients and healthy women after COVID-19 infection [16,19,25]. Solodsky et al. reported that the rate of seroconversion 15 days after documented SARS-CoV2 on RT-PCR was significantly lower in cancer patients versus healthy control women (30% versus 71%; *p* = 0.04) [25]. Palich et al. showed that almost half (45%) of cancer patients showed no anti-S antibody response after the first injection of the vaccine compared with 100% seroconversion of the healthy women, and the low seroconversion rate was much worse in elderly patients (>65 years) and patients under chemotherapy. Healthy patients not only had a 100% seroconversion rate but also had higher levels of antibody response compared with cancer patients (680 versus 315 UA/mL; *p* = 0.04) [19]. Goshen-Lago demonstrated that of the 232 patients undergoing treatment for cancer, 29% were seropositive after the first dose of vaccine compared with 84% of the controls (*p* < 0.001); however, seroconversion occurred in most cancer patients (86%) after the second dose [16].

In our study, there were no differences between the two groups in terms of antibody titers one month after the vaccination. Nonetheless, there was a more rapid trend of reduction over time among cancer patients compared with healthy women, as their titers were significantly lower after 3 months from vaccination. These findings suggest that cancer patients represent a vulnerable subgroup of the population, probably needing further booster shots to endorse their long-term protection against COVID-19 and strict serological surveillance, although the latter could be difficult given the absence of a specific cutoff able to guide vaccination frequency.

In particular, we found that cancer patients older than 50 years or obese (BMI ≥ 30 kg/m^2^) had significantly lower titers. These patients represent a particularly frail subset who could probably benefit from higher vaccine doses or anticipated booster shots to maintain their protection against COVID-19. On the other hand, there were no statistically significant correlations between seroconversion rates and FIGO stage, comorbidities, disease status (stable or in progression), or cancer treatments (ongoing or prior). This latter differs from the data reported by Palich et al., who found that active chemotherapy treatment was a factor strongly associated with no seroconversion (OR, 4.34; 95% CI 1.67–11.30; *p* = 0.003) [19].

The main strength of the study is that we included and analyzed a specific subset of the general cancer population, i.e., women presenting with gynecologic oncology malignancies. The limitations include the small sample size and the use of a specific vaccine (Pfizer-BioNTech) in a tertiary gynecologic oncology referral center.

Further large-scale prospective observational studies focusing on patients with active cancer receiving chemotherapy, targeted therapy, or immunotherapy and on cancer survivors are warranted to provide more insights on COVID-19 vaccine activity, optimal dose and frequency of booster shots, safety, and potential interactions with cancer subtypes, antineoplastic therapies, or other comorbidities.

## 5. Conclusions

The Pfizer-BioNTech COVID-19 vaccine was demonstrated to be both safe and immunogenic in gynecologic oncology patients under chemotherapy treatment. However, the antibody titer was lower in patients aged ≥ 50 years or with BMI ≥ 30 kg/m^2^. Compared with healthy women, gynecologic oncology patients showed a more rapid decline in antibody titers over time with a lower 3-month IgG titer. The data suggest that these patients would benefit from further early vaccine shots to boost their protection against COVID-19. Moreover, strict serological monitoring should be performed to assess the antibody response trend over time in this particular subpopulation.

## Figures and Tables

**Table 1 vaccines-10-00012-t001:** Demographic, clinical, and oncological data of enrolled patients.

Variable	Patients	Controls	*p* Value
	*n* = 44	*n* = 44	
Age (years) mean (SD); median (range)	61.3 (10.7); 61 (55.5–68.5)	59.5 (9.8); 60 (50–66)	0.77
Weight (kg) mean (SD); median (range)	65.4 (9.8); 64 (60.5–69)	66.6 (9.5); 65 (58–68)	0.88
BMI (kg/m^2^) mean (SD); median (range)	25.6 (5.0); 24 (19–45)	24.5 (4.8); 24 (18–42)	0.87
18–25	28 (63.6%)	30 (68.2%)	
25–30	11 (25%)	10 (22.7%)	
>30	5 (11.4%)	4 (9.1%)	
Comorbidities	33 (75%)	28 (63.6%)	0.25
Thyroid disorder	16 (36.4%)	14 (31.8%)	
Cardiovascular disease	10 (22.7%)	6 (13.6%)	
Respiratory disease	3 (6.8%)	4 (9.1%)	
Others	5 (11.4%)	4 (9.1%)	
Type of cancer			
Ovarian	28 (63.6%)		
Endometrial	9 (20.4%)		
Cervical	7 (16%)		
FIGO stage			
I	12 (27.3%)		
II	8 (18.2%)		
III	20 (45.5%)		
IV	4 (9.1%)		
Chemotherapy			
In progress	30 (68.2%)		
Within 6 months	14 (31.8%)		
Progressive disease at the time of vaccination	23 (52.3%)		

BMI, body mass index; FIGO, International Federation of Gynecology and Obstetrics.

**Table 2 vaccines-10-00012-t002:** Side effects after the first and second dose of the Pfizer-BioNTech COVID-19 vaccine.

Side Effects	Patients *n* = 44	Controls *n* = 44	*p* Value
After the second dose			
Injection site pain	20 (45.4%)	19 (43.2%)	0.83
Asthenia	15 (34%)	14 (31.8%)	0.82
Musculoskeletal pain	10 (22.7%)	11 (25%)	0.80
Headache	4 (9%)	3 (6.8%)	0.69
Nausea/vomit	1 (2.2%)	0	0.50
Fever	2 (4.4%)	2 (4.4%)	1
Chills	2 (4.4%)	2 (4.4%)	1
After the first dose			
Injection site pain	22 (50%)	20 (45.4%)	0.67
Asthenia	12 (27.3%)	10 (22.7%)	0.62
Musculoskeletal pain	8 (18.2%)	10 (22.7%)	0.60
Nausea/vomit	5 (11.4%)	3 (6.8%)	0.46
Fever	3 (6.8%)	3 (6.8%)	1
Chills	0	1 (2.2%)	0.50

**Table 3 vaccines-10-00012-t003:** Relationship between the antibody titer 1 month and 3 months after vaccination and several potential risk factors.

	Antibody Titer after 1 Month				Antibody Titer after 3 Months				
Variable	Low	Moderate	High	*p* Value	Null	Low	Moderate	High	*p* Value
	*n* = 9	*n* = 21	*n* = 14		*n* = 2	*n* = 26	*n* = 13	*n* = 3	
Age (years)				0.569					0.799
Mean (SD)	62.9 (9.9)	62.2 (9.8)	58.9 (12.8)		58.5 (3.5)	61.8 (10.1)	61.8 (10.6)	56.3 (21.5)	
Median (IQR)	58 (56–74)	62 (57–68)	58 (48.2–66.2)		58.5 (57.2–59.8)	61.5 (56.2–69.5)	64 (57–68)	46 (44–63.5)	
Weight (kg)				0.025					0.131
Mean (SD)	72.7(9.6)	63.0(10.8)	64.4 (5.8)		81 (11.3)	63.5 (8.9)	66.5 (10.6)	66.3 (4.6)	
Median (IQR)	72(65;80)	62(58;64.5)	65.5 (62; 69)		81 (77–85)	63 (58–66)	66 (62–69)	69 (65–69)	
BMI (kg/m^2^)				0.106					0.311
18–25	3 (33.3%)	14 (66.7%)	11 (78.6%)		0	16 (61.5%)	10 (76.9%)	2 (66.7%)	
25–30	3 (33.3%)	5 (23.8%)	3 (21.4%)		1 (50%)	6 (23.1%)	3 (23.1%)	1 (33.3%)	
>30	3 (33.3%)	2 (9.5%)	0		1 (50%)	4 (15.4%)	0	0	
Comorbidity	7 (77.8%)	18 (85.8%)	8 (57.1%)	0.269	1 (50%)	21 (80.8%)	10 (76.9%)	1 (33.3%)	0.269
Progressive disease	6 (66.7%)	8 (38.1%)	9 (64.3%)	0.197	2 (100%)	11 (42.3%)	8 (61.5%)	2 (66.7%)	0.313
Chemotherapy in progress	6 (66.7%)	14 (66.7%)	10 (71.4%)	0.951	2 (100%)	18 (69.2%)	9 (69.2%)	1 (33.3%)	0.452
FIGO stage				0.69					0.281
I	1 (11.1%)	9 (42.8%)	2 (14.3%)		1 (50%)	6 (23.1%)	3 (23.1%)	2 (66.7%)	
II	2 (22.2%)	3 (14.3%)	3 (21.4%)		0	4 (15.4%	4 (30.8%)	0	
III	4 (44.4%)	8 (38.1%)	8 (57.1%)		0	13 (50%)	6 (46.2%)	1 (33.3%)	
IV	2 (22.2%)	1 (4.8%)	1 (7.1%)		1 (50%)	3 (11.5%)	0	0	

BMI, body mass index; FIGO, International Federation of Gynecology and Obstetrics; IQR, interquartile range; SD, standard deviation.

**Table 4 vaccines-10-00012-t004:** Antibody titer of oncologic patients according to several variables.

Variable	Antibody Titer (Median, IQR)			
	After 1 Month	*p* Value	After 3 Months	*p* Value
Age (years)		0.018		0.004
<50	1530 (885–2605)		697 (452–840)	
≥50	634 (298–844)		194 (115–373)	
BMI (kg/m^2^)		0.016		0.156
<30	759 (406–1947)		232 (162–558)	
≥30	264 (180–578)		170 (107–214)	
Comorbidities		0.111		0.147
Yes	604 (322–780)		191 (120–400)	
No	1060 (799–1925)		342 (203–697)	
Progressive disease		0.592		0.681
Yes	726 (374–2340)		226 (104–669)	
No	680 (353–917)		197 (169-364)	
Chemotherapy in progress		0.871		0.186
Yes	734 (361–1675)		203 (102–426)	
No	642 (332–1048)		340 (171–612)	
FIGO stage		0.507		0.181
I	687 (549–814)		322 (200–464)	
II	546 (265–1365)		285 (125–562)	
III	782 (380–2287)		203 (167–608)	
IV	422 (241–641)		86 (55–129)	

BMI, body mass index; IQR, interquartile range.

**Table 5 vaccines-10-00012-t005:** Antibody titer of oncologic patients compared with healthy vaccinated women.

Antibody Titer	Patients	Controls	*p* Value
	*n* = 44	*n* = 44	
After 1 month			0.101
Mean (SD)	1440.8 (2216.6)	1804.3 (2447.0)	
Median (IQR)	710 (345–1287)	783 (647–1322)	
After 3 months			<0.001
Mean (SD)	482.8 (969.1)	992.6 (1177.5)	
Median (IQR)	214 (154–525)	550 (370–781)	

IQR, interquartile range; SD, standard deviation.

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
