# Peer review of "Pfizer-BioNTech COVID-19 Vaccine in Gynecologic Oncology Patients: A Prospective Cohort Study"

_vaccines, 2021, doi:10.3390/vaccines10010012_

Round 1
Reviewer 1 Report
The manuscript entitled Safety and immunogenicity of the Pfizer-BioNTech COVID-19 vaccine in gynecologic patients under chemotherapy treatment: a prospective cohort study by Innocenza Palaia et al. for the first time reports data of a small single-center study regarding safety and immunologic efficacy of Corona vaccination in a group of gynecologic cancer patients under therapy.
All in all, the study is well performed and regards many factors influencing an immune response and side effects of a vaccination. Some minor issues have to be clarified.
- The title could be misleading. Did all patients still receive chemotherapy at the time of vaccination? The sentence (lines 71-74) “Key inclusion criteria….(c) chemotherapy treatment in progress or completed within 6 months.” is not precise. Does this mean that some of the patients received therapy up to 6 months ago or does it mean that therapy will be completed within the next 6 months? Please specify your statement and correct the title if necessary.
- The control group is not well characterized (there is no information about age, BMI, diseases, other medications). In particular, the age match to the cohort of patients is a prerequisite in order to compare the two groups.
- Table 5 (line 150): there is a mistake regarding the BMI data. Within the text (line 157) the authors state that “…patients with BMI<30 kg/m2 had a higher 1-month antibody titer compared to those with BMI ≥30 kg/m2…”. The table presents contradictory data.
- Please unify the spelling of the vaccine to Pfizer-BioNTech COVID-19 vaccine.
- The authors’ statement that all patients, regardless of their antibody titer, are protected against an infection with SARS-CoV-2 is incorrect. Please rephrase the sentence.
Author Response
REVIEWER 1
The manuscript entitled Safety and immunogenicity of the Pfizer-BioNTech COVID-19 vaccine in gynecologic patients under chemotherapy treatment: a prospective cohort study by Innocenza Palaia et al. for the first time reports data of a small single-center study regarding safety and immunologic efficacy of Corona vaccination in a group of gynecologic cancer patients under therapy.
All in all, the study is well performed and regards many factors influencing an immune response and side effects of a vaccination. Some minor issues have to be clarified.
Comment 1:
The title could be misleading. Did all patients still receive chemotherapy at the time of vaccination?
Answer 1:
Thank you for the clarification. Some patients were under chemotherapy treatment while others completed chemotherapy within 6 months before being vaccinated. In order to avoid any misunderstanding, we decided to change the title of the manuscript to:
“Safety and immunogenicity of the Pfizer-BioNTech COVID-19 vaccine in gynecologic oncology patients under chemotherapy treatment: A prospective cohort study”
We clarified the specific population setting in the appropriate sections, i.e., Abstract and Methods:
- Lines 16-17. “Abstract […] Methods: […] who were under chemotherapy or had completed it within 6 months at the time of the study. were All patients received a two-dose schedule of the Pfizer-BioNTech COVID-19 vaccine. Results were compared with a control group of healthy women vaccinated in the same period.”
- Lines 76-77. “Key inclusion criteria […] (c) patients under chemotherapy treatment or who had completed chemotherapy within the prior 6 months; […].”.
Comment 2:
The sentence (lines 71-74) “Key inclusion criteria…(c) chemotherapy treatment in progress or completed within 6 months.” is not precise. Does this mean that some of the patients received therapy up to 6 months ago or does it mean that therapy will be completed within the next 6 months? Please specify your statement and correct the title if necessary.
Answer 2:
As stated in the prior Answer 1, we clarified this aspect as requested (lines 76-77):
“Key inclusion criteria […] (c) patients under chemotherapy treatment or who had completed chemotherapy within the prior 6 months; […].”.
Comment 3:
The control group is not well characterized (there is no information about age, BMI, diseases, other medications). In particular, the age match to the cohort of patients is a prerequisite in order to compare the two groups.
Answer 3:
Thank you for the comment. We asked through written informed consent to access the demographic data of the health care workers selected for the study, which were recorded in our Department of Molecular Medicine, Sapienza University. We added the demographic data (age, weight, BMI, comorbidities) requested in Table 1. There were no statistically significant differences in terms of demographic data between cancer patients and controls and we added this concept in lines 119-121.
“There were no statistically significant differences in terms of demographic data between cancer patients and controls.”
Comment 4:
Table 5 (line 150): there is a mistake regarding the BMI data. Within the text (line 157) the authors state that “…patients with BMI <30 kg/m2 had a higher 1-month antibody titer compared to those with BMI ≥30 kg/m2…”. The table presents contradictory data.
Answer 4:
Thank you for the comment. We made a mistake and inverted the lines in the table. We corrected the data in the ex Table 5 (now Table 4; see other revisions) accordingly.
Comment 5:
Please unify the spelling of the vaccine to Pfizer-BioNTech COVID-19 vaccine.
Answer 5:
Thank you for pointing it out. There was a typo in line 57. We corrected it accordingly.
Comment 6:
The authors’ statement that all patients, regardless of their antibody titer, are protected against an infection with SARS-CoV-2 is incorrect. Please rephrase the sentence.
Answer 6:
Thank you for your suggestion, thus we changed the sentence in lines 229-230 as follows:
“All our patients experienced an adequate seroconversion and none of them had COVID-19 infection during the study., suggesting that their antibody titers, albeit different, were able to protect them against the virus.”.

Reviewer 2 Report
Some important previous related studies not mentioned in the introduction and in discussion. e.g.:
1.So, A.C.P.; McGrath, H.; Ting, J.; Srikandarajah, K.; Germanou, S.; Moss, C.; Russell, B.; Monroy-Iglesias, M.; Dolly, S.; Irshad, S.; Van Hemelrijck, M.; Enting, D. COVID-19 Vaccine Safety in Cancer Patients: A Single Centre Experience. Cancers 2021, 13, 3573. https://doi.org/10.3390/cancers13143573
2.Forster M, Wuerstlein R, Koenig A, et al. COVID-19 vaccination in patients with breast cancer and gynecological malignancies: A German perspective [published online ahead of print, 2021 Oct 29]. Breast. 2021;60:214-222. doi:10.1016/j.breast.2021.10.012
Decision: Major revision, Reconsider submission after more accurate review of literature to improve the manuscript, specifically the discussion and conclusion.
Author Response
REVIEWER 2
Comment:
Some important previous related studies not mentioned in the introduction and in discussion. e.g.:
- So, A.C.P.; McGrath, H.; Ting, J.; Srikandarajah, K.; Germanou, S.; Moss, C.; Russell, B.; Monroy-Iglesias, M.; Dolly, S.; Irshad, S.; Van Hemelrijck, M.; Enting, D. COVID-19 Vaccine Safety in Cancer Patients: A Single Centre Experience. Cancers 2021, 13, 3573. https://doi.org/10.3390/cancers13143573
- Forster M, Wuerstlein R, Koenig A, et al. COVID-19 vaccination in patients with breast cancer and gynecological malignancies: A German perspective [published online ahead of print, 2021 Oct 29]. Breast. 2021;60:214-222. doi:10.1016/j.breast.2021.10.012
Answer:
Thank you for the interesting suggestions.
We updated the review of the literature and we included in our manuscript the suggested references both in Introduction and Discussion. Moreover, we added a paragraph in the Discussion section to report the data of the study by Forster et al., evaluating a German perspective on COVID-19 vaccination in patients with breast and gynecological malignancies (lines 220-222):
“[…] Recently, Forster et al. published a German study reporting that COVID-19 vaccination was well-tolerated by patients with breast and gynecological cancer undergoing systemic cancer therapy [18] […].”

Reviewer 3 Report
In this article researchers report their attempt of evaluating safety and immunogenicity of Pfizer-BioNTech COVID-19 vaccine in gynecologic oncology patients. This experiment is not properly planned and executed, is not scientifically rigorous, lacks required methods, results are not interpreted intellectually and overall, the manuscript of very poor scientific quality.
Major comments:
- The manuscript has two aspects, ‘safety’ and ‘immunogenicity’, but safety part is completely missed. It is not clear how did they assess safety of the vaccine in their cohort. The local and systemic events both after first and second dose of vaccination in both cohorts (cancer patients and healthy controls) must be considered and compared over a certain period (e.g., 7 days or 30 days). The authors can use the earlier works on COVID-19 vaccine safety and immunogenicity as reference to understand why their work is not of scientific merit to report ‘safety and immunogenicity’.
- Frenck et al., NEJM. https://www.nejm.org/doi/full/10.1056/NEJMoa2107456
- Anderson et al., NEJM. https://www.nejm.org/doi/full/10.1056/NEJMoa2028436
- The inclusion of healthy control is mentioned but their demographic, safety and other data are not compared side by side except for the comparison of antibody responses.
- The pre-existing immunity due to earlier infection, both in healthy controls and cancer patients, are of prime importance. Without such screening and information, subsequent data will not have appropriate meaning.
- This experiment does not mention anything about ethical approval. Also does not have any funding sources mentioned that raised concern over ethics and integrity.
- Objectives are not fulfilled. The authors mention that they would like to compare satety and immunogenicity by cancer types, disease status, treatment status etc. but neither of these are evaluated systemically.
- Result section does not have any intellectual engagement. Just showing data on spaghetti plot or any other figures does not tell anything. Same results are presented in different tables/figures. E.g., tables 3 and 4 can be merged into single table; table 5 should have several other variables which fit well with the objective of the paper including cancer types, treatment status etc., comparison of healthy controls vs cancer patients is missing everywhere, figure 1 does not make any sense to the essence of the paper, table 6 and figure 2 show same information.
- Manuscript written very poorly.
- In introduction, they mention ‘elevated number of cancer patient’ but provide no evidence. They also mention having no data of vaccine safety and immunogenicity in cancer patients and mention immediately that there is no concern over vaccine safety. Instead, data on safety and immunogenicity in cancer patients are being published regularly recently.
- In methods, it is not clear what do they mean by ‘no clinical contradictions to COVID-19 vaccination’; in lines 84-86, not clear what the authors would like to state.
- In results: what is FIGO state, they have not mentioned anywhere. The sum of patients by ‘type of cancer’ is not 44.
- Discussion: The first and second paragraphs of the discussion are basically same information from introduction, nothing extra. Lines 212-213 mentions that vaccine is well-tolerated by both cohorts, but such data are not provided.
- Conclusion: Again, conclusion has same thing as in discussion.
Overall, the introduction, results, discussion, and conclusion do not involve much scientific engagements and repeat same information again and again.
Author Response
REVIEWER 3
Comment 1:
The manuscript has two aspects, ‘safety’ and ‘immunogenicity’, but safety part is completely missed. It is not clear how did they assess safety of the vaccine in their cohort. The local and systemic events both after first and second dose of vaccination in both cohorts (cancer patients and healthy controls) must be considered and compared over a certain period (e.g., 7 days or 30 days).
Answer 1:
We would like to thank the reviewer for careful and thorough reading of this manuscript and for the constructive suggestions, which helped to improve its quality.
We specified in the Methods that we recorded adverse events both after first and second dose of vaccination in both groups over a 7-day period. We enriched Table 2 by adding also the side effects after the first dose and the events recorded in the control group. No statistically significant differences were reported between the two groups.
We also modified the manuscript in lines 129-135.
“None of the patients experienced serious adverse effects after vaccination. Both after the first and the second dose, the most common side effects were the pain at the site of injection, which was reported in 20 (45.4%) patients and disappeared within a few days, followed by asthenia in 15 (34%) patients, and musculoskeletal pain in 10 (22.7%). Two (4.4%) patients had fever (≥38 °C) after vaccination requiring antipyretics. No statistically significant differences in terms of post-vaccine shot adverse events were reported between cancer patients and healthy controls.”
Comment 2:
The authors can use the earlier works on COVID-19 vaccine safety and immunogenicity as reference to understand why their work is not of scientific merit to report ‘safety and immunogenicity’. Frenck et al., NEJM. https://www.nejm.org/doi/full/10.1056/NEJMoa2107456 Anderson et al., NEJM. https://www.nejm.org/doi/full/10.1056/NEJMoa2028436
Answer 2:
Thank you for the comment. In order to avoid any misunderstanding or presumption, we decided to change the title of the manuscript to: “Safety and immunogenicity of the Pfizer-BioNTech COVID-19 vaccine in gynecologic oncology patients under chemotherapy treatment: A prospective cohort study”.
Comment 3:
The inclusion of healthy control is mentioned but their demographic, safety and other data are not compared side by side except for the comparison of antibody responses.
Answer 3:
Thank you for the comment. We kindly asked through written informed consent to access to the demographic data of the health care workers selected for the study, which were recorded in our Department of Molecular Medicine, Sapienza University We added the demographic data (age, weight, BMI, comorbidities) in Table 1. There were no statistically significant differences in terms of demographic data between cancer patients and controls and we added this concept in lines 119-121.
Comment 4:
The pre-existing immunity due to earlier infection, both in healthy controls and cancer patients, are of prime importance. Without such screening and information, subsequent data will not have appropriate meaning.
Answer 4:
Thank you for the comment. As requested, we added the baseline IgG titers of included patients before vaccination. All patients had negative titers and we added this meaningful information in our Methods (lines 79-80):
“[…] Key inclusion criteria included: […] (e) negative baseline SARS-CoV-2 IgG antibodies before vaccination […]”
Comment 5:
This experiment does not mention anything about ethical approval. Also does not have any funding sources mentioned that raised concern over ethics and integrity.
Answer 5:
We already discussed this important point with the Assistant Editor. We consulted our local Institutional Review Board (Policlinico Umberto I of Rome) on October 20, 2020 before starting to collect data and we uploaded the report (in Italian language) in the submission system.
Comment 6:
Objectives are not fulfilled. The authors mention that they would like to compare safety and immunogenicity by cancer types, disease status, treatment status etc. but neither of these are evaluated systemically.
Answer 6:
Thank you for the comment. We enriched the new Table 4 by adding the comparison of antibody titers of oncologic patients according to other potential risk factors (progressive disease, chemotherapy in progress, FIGO stage).
Comment 7:
Result section does not have any intellectual engagement. Just showing data on spaghetti plot or any other figures does not tell anything. Same results are presented in different tables/figures. E.g., tables 3 and 4 can be merged into single table; table 5 should have several other variables which fit well with the objective of the paper including cancer types, treatment status etc., comparison of healthy controls vs cancer patients is missing everywhere, figure 1 does not make any sense to the essence of the paper, table 6 and figure 2 show same information.
Answer 7:
Thank you for the suggestions.
- We merged Table 3 and Table 4 into a single table (new Table 3).
- We enriched ex Table 5 (now Table 4) with other variables (progressive disease, chemotherapy in progress, FIGO stage) as requested.
- We removed Figure 1 as suggested.
- We decided to maintain ex Figure 2 (now Figure 1) as it gives a more representative overview of the data detailed in ex Table 6 (now Table 5).
Comment 8:
Manuscript written very poorly. In introduction, they mention ‘elevated number of cancer patient’ but provide no evidence.
Answer 8:
We provided the updated estimates of the GLOBOCAN database 2020 to quantify the global cancer burden in 2020 [Reference 1]:
“Sung H, Ferlay J, Siegel RL, Laversanne M, Soerjomataram I, Jemal A, et al. Global Cancer Statistics 2020: GLOBOCAN Estimates of Incidence and Mortality Worldwide for 36 Cancers in 185 Countries. CA Cancer J Clin 2021;71:209–49. https://doi.org/10.3322/caac.21660.”.
Comment 8:
They also mention having no data of vaccine safety and immunogenicity in cancer patients and mention immediately that there is no concern over vaccine safety. Instead, data on safety and immunogenicity in cancer patients are being published regularly recently.
Answer 8:
Thank you for the comment. We modified the sentence in order to avoid any misunderstanding and enriched data by adding two interesting references [17,18] as follows (line 45-46):
“[…] However, till today there is not enough data to prove […]”.
Moreover, we added a paragraph in the Discussion section to report the data of the study by Forster et al., evaluating a German perspective on COVID-19 vaccination in patients with breast and gynecological malignancies (lines 220-222):
“[…] Recently, Forster et al. published a German study reporting that COVID-19 vaccination was well-tolerated by patients with breast and gynecological cancer undergoing systemic cancer therapy [18] […].”
Comment 9:
In methods, it is not clear what do they mean by ‘no clinical contradictions to COVID-19 vaccination’; in lines 84-86, not clear what the authors would like to state.
Answer 9:
We clarified the meaning of this exclusion criteria and modified the manuscript accordingly (lines 78-79):
“(d) no clinical contraindications to COVID-19 vaccination (e.g., severe allergic reaction to a previous dose or a known allergy to a component of the vaccine)”.
Comment 10:
In results: what is FIGO state, they have not mentioned anywhere.
Answer 10:
Thank you for pointing it out. We explicated the meaning of FIGO (International Federation of Gynecology and Obstetrics) staging (lines 114-115).
Comment 11:
The sum of patients by ‘type of cancer’ is not 44.
Answer 11:
Thank you for pointing it out. There was a typo in Table 1. The number of cervical cancer patients was 7 and not 4 (as correctly stated in the Abstract and in the Results). We corrected the Table 1.
Comment 12:
Discussion: The first and second paragraphs of the discussion are basically same information from introduction, nothing extra.
Answer 12:
Thank you for the comment. We revised the first two paragraph of the Discussion section and removed some redundant concepts, already explained in the Introduction (lines 197-210):
“In the context of the ongoing global COVID-19 pandemic, caused by severe acute respiratory syndrome coronavirus 2 (SARS-CoV-2), the preventive role of vaccines be-comes even more important for immunocompromised individuals, such as cancer patients, who demonstrated to be at higher risk of severe complications, intensive care unit (ICU) admissions, and death [6,7,24]. Vaccines against SARS-CoV-2 act by eliciting the production of antibodies against the viral surface spike (S) protein, thus blocking the viral entry into host cells [25,26]. Currently, they represent the only preventative measure against this life-threatening global infection along with hygiene precautionary measures [8].
However, there is only scant evidence specifically addressing the immunogenicity and safety of available vaccines in cancer patients, since immunocompromised individuals were excluded from initial registration trials. Cancer patients have an immunocompromised status both because of the disease itself and the cancer treatments used; therefore, it is plausible that they present lower rates of antibody seroconversion and so a higher risk of severe infections [10,13,27,28]. […]”.
Comment 13:
Lines 212-213 mentions that vaccine is well-tolerated by both cohorts, but such data are not provided.
Answer 13:
As stated in Answer 1, we enriched Table 2 by adding the side effects after both the first dose and the second dose and by adding the adverse events recorded also in the control group.
Comment 14:
Conclusion: Again, conclusion has same thing as in discussion. Overall, the introduction, results, discussion, and conclusion do not involve much scientific engagements and repeat same information again and again.
Answer 14:
Thank you for the comment. We revised the Conclusion section and removed redundant sentences (lines 275-277):
“[…] None of the patients experienced serious adverse effects. All patients receiving the two-dose regimen vaccine presented an adequate seroconversion. […]”.

Round 2
Reviewer 2 Report
In table 1 and 2, p values shall be reported with 3 decimals, not just reporting NS
Decision: Accept, with minor revision.
Author Response
REVIEWER 2
Comment:
In table 1 and 2, p values shall be reported with 3 decimals, not just reporting NS.
Answer:
Thank you for the comment. We reported the p values as requested.

Reviewer 3 Report
Authors have addressed the earlier comments. Extensive English language edit is still necessary. The authors need to keep either table 5 or figure 1. If they choose figure 1, they need to include the statistics in figure, as it looks like at 3 month there is signifiant difference. The Spaghetti plot in figure 2 is not necessary as it does not provide any additional information like the rate of antibody decay/persistance, antibody half-life etc.
Author Response
REVIEWER 3
Comment:
Authors have addressed the earlier comments. Extensive English language edit is still necessary. The authors need to keep either table 5 or figure 1. If they choose figure 1, they need to include the statistics in figure, as it looks like at 3 month there is significant difference. The Spaghetti plot in figure 2 is not necessary as it does not provide any additional information like the rate of antibody decay/persistence, antibody half-life etc.
Answer:
Thank you for the comment. The paper has been carefully revised by a native English speaker. We removed Figure 1 and Figure 2 as suggested.

Round 3
Reviewer 3 Report
Comments are addressed.